# Role of Neutrophil Extracellular Traps in Health and Disease Pathophysiology: Recent Insights and Advances

**DOI:** 10.3390/ijms242115805

**Published:** 2023-10-31

**Authors:** Md. Monirul Islam, Naoshi Takeyama

**Affiliations:** 1Department of Emergency and Critical Care Medicine, Aichi Medical University, Aichi 480-1195, Japan; shafinbiochemistry@gmail.com; 2Department of Biochemistry and Biotechnology, University of Science and Technology Chittagong (USTC), Chattogram 4202, Bangladesh

**Keywords:** neutrophil extracellular traps (NETs), innate immunity, sepsis, lung disease, cardiovascular disease, liver disease, kidney disease, diabetes, COVID-19, coagulopathy and thrombotic microangiopathy, cancer, autoimmunity, preeclampsia, kawasaki disease

## Abstract

Neutrophils are the principal trouper of the innate immune system. Activated neutrophils undergo a noble cell death termed NETosis and release a mesh-like structure called neutrophil extracellular traps (NETs) as a part of their defensive strategy against microbial pathogen attack. This web-like architecture includes a DNA backbone embedded with antimicrobial proteins like myeloperoxidase (MPO), neutrophil elastase (NE), histones and deploys in the entrapment and clearance of encountered pathogens. Thus NETs play an inevitable beneficial role in the host’s protection. However, recent accumulated evidence shows that dysregulated and enhanced NET formation has various pathological aspects including the promotion of sepsis, pulmonary, cardiovascular, hepatic, nephrological, thrombotic, autoimmune, pregnancy, and cancer diseases, and the list is increasing gradually. In this review, we summarize the NET-mediated pathophysiology of different diseases and focus on some updated potential therapeutic approaches against NETs.

## 1. Introduction to Neutrophil Extracellular Traps (NETs)

### 1.1. Neutrophil and Innate Immunity

Innate immunity is the first line of defense of the body. The main function of the innate immunity is to protect our body from invading microorganisms and limit the growth and proliferation and finally kill the microorganisms. Neutrophils are the most abundant granular leukocytes produced at a rate of 5 × 10^10^–10 × 10^10^ cells per day, contributing approximately 50–70% of all circulating white blood cells in humans, having an average short half-life of 6–8 h in circulation and are key players of the innate immunity [1,2]. They are the first responder of the inflammatory cascades [3,4]. Neutrophils are derived from the bone marrow and enter into the circulation from where they quickly move into the sites of infection or inflammation in response to pathogen attack in the body and roll, adhere to endothelial layer, crawl, and thus transmigrate from the vessel and kill the microbes [5,6,7]. After accomplishment of their role, they undergo apoptosis and are cleared by macrophages [8]. As the arsenal of innate immunity, neutrophils appears first at the infected site in the body and imply several strategies to eliminate the infection. In fact, neutrophils evolved to fulfill the key role in innate immunity through rapid deployment and effective antimicrobial action against a broad range of pathogens and noninfectious inflammation. Hence, they are equipped with multiple weapons that can be deployed as a part of their antimicrobial strategies [9].

### 1.2. Antimicrobial Actions of Neutrophils

Neutrophils were recognized as the vanguard of the innate immunity and a vital protector against microbial infection and foreign invasion. In past decades, it was thought that when neutrophils encounter invading pathogens in the body, they kill them either by phagocytosis, where they engulf and inactivate the pathogen and rapidly kill them by releasing proteolytic enzymes, antimicrobial proteins or reactive oxygen species (ROS) or by exocytosis, where they degranulate and release antimicrobial factors in the extracellular space. However, in 2004, Brinkmann et al. described another way of neutrophil-derived novel antimicrobial defense mechanism of innate immunity termed NETosis that can participate in pathogen killing extracellularly through the release of NETs [10]. Although neutrophils are classically known to die primarily by apoptosis or necrosis and highly regulated necroptosis, NETosis is a cell death program that is distinct from apoptosis and necrosis [11,12].

### 1.3. Neutrophil Extracellular Traps (NETs)

NETs are released by activated neutrophils, and the structure is composed of a meshwork of decondensed chromatin fibers decorated with antimicrobial granular proteins such as histones, myeloperoxidase (MPO), neutrophil elastase (NE), and cathepsin G. This weblike architectural design and composition allow NETs to prevent the dissemination of pathogen in the body. NETs can only be released robustly by matured neutrophils upon stimulation. Immature neutrophils are less potent compared to mature neutrophils at promoting innate immune defenses [13,14,15], and neutrophils from term and preterm infants fail to form NETs upon activation by inflammatory agonist [16]. Several infectious and sterile stimuli have been reported to trigger NET formation including bacteria [10,12], viruses [17], fungi [18,19], parasites [20], pro-inflammatory cytokines like interleukin 8 (IL-8) [21], tumor necrosis factor α (TNF-α) [22], placental micro-debris [23], activated platelets [24], cholesterol crystals, monosodium uric acid [25], immune complexes [26], autoantibodies [27], complements such as C5a [28], and even cancer cells [29,30,31]. Phorbol esters such as phorbol myristate acetate (PMA) and bacterial products such as lipopolysaccharide (LPS) are the most widely used nonphysiological agonists to generate NETs ex vivo [10,32,33].

### 1.4. NETosis Mechanism

Two different mechanisms, NADPH Oxidase 2 (Nox 2)-dependent and Nox 2-independent NETosis have been well validated by published literature (Figure 1). Agonists like PMA, LPS, and bacteria such as *Pseudomonas aeruginosa* induce Nox-dependent NETosis, while agonists like calcium ionophores (A231128, ionomycin), uric acid crystals, certain microbes, and UV light trigger Nox-independent NETosis through the formation of different ROS; Nox-ROS and mitochondrial ROS, respectively [34,35,36]. The generated ROS activates Nox-dependent and Nox-independent specific different sets of kinases (MAPK, ERK, p38, and JNK) leading to transcriptional firing and stimulation of MPO. In Nox-independent NETosis, histone citrullination is facilitated by a nuclear enzyme protein arginine deiminase 4 (PAD4) activation [37,38]. MPO triggers the stimulation and translocation of NE from azurophilic granules to the nucleus and decorates the chromatin [39,40,41]. Ultimately, nuclear membrane disintegrates, and NETs are released.

### 1.5. NETosis Pathways

NET formation occurs via two pathways: suicidal NETosis and vital NETosis. Suicidal NETosis represents a cell death pathway that starts with nuclear delobulation, the nuclear membrane disassembly followed by continuous loss of cellular polarization, chromatin decondensation, and finally finished with plasma membrane rupture and expulsion of NETs [42]. This lytic slow cell death process usually takes 2–4 h [33]. On the other hand vital, NETosis is a cell-death-independent non-lytic process whereby expulsion of nuclear chromatin is accompanied by the release of granular proteins, and fabrication of these components occurs extracellularly, leaving active anucleated phagocytic cytoplast having the potency to ingest microorganisms and chemotaxis [42]. Vital NETosis happens faster, within 5–60 min depending on the inducer [43]. In vital NETosis, the released extracellular DNA by neutrophils could be either nuclear or mitochondrial [44,45,46].

### 1.6. Physiological Role of NETs

Decades ago, when the NETs emerged, scientists became very interested in observing its antibacterial capacity. The wide range of killing capacities of NETs made the immunologist work with great haste. NETs conquer infections by trapping, immobilizing, and neutralizing Gram-negative and Gram-positive bacteria [10], viruses [17] fungi [47], parasites [48,49], and thus they prevent the dissemination of intruding microbes and protect the host (Figure 2). Although NETs trap microbes through charge interaction [50], some pathogens evade NETs. Many Gram-positive pathogens, including but not limited to pathogens of the *Streptococcus* and *Staphylococcus* genera, release endonucleases, and secreted endonuclease degrades the extracellular DNA scaffold, destroying and evading NETs [51,52]. The escaped bacteria with endonuclease may promote further invasion and spread from the local sites to distant organs and the bloodstream [53,54,55]. Again, *Streptococcus pneumoniae* can escape from NETs in a charge-dependent manner. This Gram-positive bacterial strain expresses anti-phagocytic polysaccharide capsules and lipoteichoic acid which can produce a positive charge on their surface and thus prevent them from being trapped by positively charged NET fibers and histone residues [56]. Interestingly, the beneficial effect of NETosis over phagocytosis is the size of NETs. Large microbes and parasites evade phagocytosis and can prove difficult to clear. But neutrophil has a unique microbe-sensing mechanism that allows them to selectively tailor their antimicrobial responses to pathogen on the basis of microbial size. The large filamentous form of fungi cannot be removed by phagocytosis. NETs play a significant role in controlling these large filamentous fungi [57,58,59].

In addition, NETs also contribute to immunothrombosis. Immunothrombosis is a physiological process having a bidirectional role with the innate immune system. Although excessive activation of coagulation cascade leads to many clinical conditions including sepsis, disseminated intravascular coagulation (DIC), myocardial infarction (MI), and coronavirus disease 2019 (COVID-19) [60], in contrast, it also plays an essential protective role in maintaining physiological hemostasis to avoid blood loss and arresting both viral and bacterial infections. However, with the discovery of NETs, it has come to light that thrombus is not only for the hemostatic purpose of stopping bleeding but also takes part in innate immunity. NETs provide a large procoagulant surface by activating the contact phase of coagulation [61,62]. Moreover, neutrophils and neutrophils-derived micro-particles during NETosis such as DNA, histones, and granule proteins provide coagulant activities [61].

### 1.7. Controversial Role of NETs

In fact, the intervention of NETs is a landmark progression of science. Now the question is whether NETs are too good or too bad for us. NETs are like a double-edged sword. While NETs have a physiological role in antimicrobial defenses, if dysregulated, they also have various pathological aspects that have attracted recent attention (Figure 3). In some conditions when it is excessively generated and present, NETs can do harm to the host. The antimicrobial histones and peptides decorating the NET-DNA are directly cytotoxic to tissue and ineffective clearance of NETs results in deleterious inflammation of the host tissue. The list of disorders implicated by NETs includes sepsis, pulmonary, cardiovascular, hepatic, nephrological, thrombotic, autoimmune, pregnancy, and cancer diseases and are gradually increasing.

### 1.8. Detection of NETs

Uncontrolled release of NETs has been reported in several disease pathophysiology (discussed later). Therefore, precise spotting of NETs in clinical samples could have great potential in the prognosis of disease progression and subsequent consequences. NETs are made up of cell-free DNA (cf-DNA) and antimicrobial proteins like MPO, and NE NET detection is very challenging due to its fragile structure, timing of formation and turnover, and frequent presence of DNase. Measuring cf-DNA is not a good approach to quantify NETs as the source could also be apoptosis and necrosis in addition to NETosis [63]. NETs can be visualized and quantified by fluorescence microscopy [64] and flow cytometry [65] tools in human neutrophils in vitro. However, enzyme-linked immunosorbent assay is the handy, highly sensitive, and reliable method to quantitatively measure remnants of circulating NETs in plasma [66,67], as well as in cell culture supernatants in vitro [68].

## 2. NETs in Clinical Settings

### 2.1. NETs in Sepsis

Sepsis is a dysregulated response to an infection with deleterious effects in a host, leading to circulatory shock, multiple organ failure syndromes (MODS), and ultimately death. A common cause of death in sepsis is the overwhelming infection in the bloodstream and the resulting complications [69]. During sepsis, activation of neutrophil with microbial or inflammatory stimuli occurs, which results in the expulsion of NETs [70]. NETs are an essential antimicrobial defense for pathogen clearance in the blood and tissues during infection, but at the same time, NETs and NET components exert excessive inflammation, resulting host tissue damage [42]. Using an animal model of sepsis, circulatory NET in the bloodstream became evident [24,45,71], and the biomarkers used to check the presence of NETs were also increased in septic patients [72,73,74]. NET damages tissue and increases vascular permeability in sepsis. Promotion of neutrophil infiltration occurs in tissues through neutrophil–endothelial cell (EC) interaction [75]. This interaction results in excess NET formation which is dependent on activated EC-derived IL-8 [76]. This damage is neutralized when incubated with either NAPDH oxidase inhibitors or DNase [76]. 

Immunothrombosis represents controlled inflammation and coagulation and is the major line of innate immune defense against intruding infection. In sepsis, dysregulated, sustained, and hyper-immunothrombosis leads to DIC complications [77]. Sepsis-induced DIC is detrimental to the host and causes organ dysfunctions, having a mortality rate double that of septic patients without DIC [78]. Neutrophils diligently participate in thrombosis-associated DIC [79]. Neutrophil-released NETs contribute excessive thrombin generation due to their early presence at the onset of DIC [80].

Normally, platelet remains dormant in the circulation [81]. A growing body of evidence reveals that, in sepsis, the interaction between neutrophils and activated platelets happens simultaneously, and that platelets can rapidly mediate neutrophils to make NET in vivo [24,82]. Activated platelets trigger the neutrophils-derived NET release in either a P-selectin-dependent [83,84] or Toll-like receptor (TLR) 4-dependent manner [24,85], which traps bacteria at the expense of endothelial tissue damage. Histones along with fragmented DNA are the abundant components of NETs. In acute systemic inflammatory conditions, including sepsis and trauma, circulating histones aggravate micro-circulatory thrombosis, worsen tissue perfusion, and contribute significantly to organ injury [85].

### 2.2. NETs in Lung Diseases

Excessive activation of neutrophils causes MODS, and the lung, the most sensitive and important organ in systemic inflammation, is the main target [86,87]. Acute lung injury (ALI) is one of the leading causes of death in the ICU. Lung edema, inflammation, hyaline membrane, and alveolar damage are the characteristic morphological features of ALI [88]. Acute respiratory distress syndrome (ARDS) is an acute inflammatory lung injury characterized by hypoxemic respiratory failure as a consequence of increased permeability of the endothelial–epithelial barrier, alveolar damage, and pulmonary edema. ALI and its severest form ARDS, or chronic obstructive pulmonary disease, remains an important clinical challenge due to its complex and ambiguous pathophysiology [89,90]. A massive influx of activated neutrophils is seen to the lung microvasculature, interstitial, and alveolar space, and dysregulated inflammatory neutrophils are the key factor in the progression of ALI/ARDS [91,92,93]. This excessive neutrophil activation and accumulation induces increased formation of NETs along with increased release of proinflammatory mediators [94,95]. NET-mediated cytotoxicity on alveolar epithelial cells as well as pulmonary EC is mainly due to protein components of NETs [96]. Histones, major components of NETs, are too toxic to cells and the lung is the most susceptible vital organ to high levels of circulatory histones [97]. NET-derived histones were detected from bronchoalveolar lavage fluid (BALF) samples from humans with ARDS [98]. Co-culture of PMA-stimulated neutrophil with EC alters barrier function, resulting in EC damage, which is attributed to NETs and pretreating EC with DNase, Cl-amidine, a PAD4 inhibitor, and 1-(3-methylbenzoyl)-1H-indazole-3-carbonitrile, an NE inhibitor, restore the damage [99]. A growing body of evidence suggested that the lung injury in ALI or ARDS is triggered by C5a-activated NET release along with histones and enzymes that cause tissue damage [98,100]. NETs also contribute to pathogen-induced lung injury in mouse models and human [101].

Cystic fibrosis (CF) is an inherited autosomal recessive disease of the lung characterized by chronic inflammation, bacterial colonization, and mucus overproduction in the airways leading to morbidity and mortality in patients [102,103]. CF occurs due to a mutation in the CF trans-membrane conductance regulator (*CFTR*) gene that encodes CFTR protein [103]. This anion channel regulates the balance of bicarbonate and chloride secretions across the cell surface epithelium of the airways [102]. This mutation increases the susceptibility of these patients to airway bacterial infection mainly by *Pseudomonas aeruginosa* and *Haemophilus influenza* and *Staphylococcus aureus* [104]. Neutrophils are infiltrated into the airway upon bacterial colonization and show less potency to eliminate microbes, instead contributing to lung damage [105]. This colonization induces NET expulsion, resulting in sputum viscosity and ultimately exacerbating the patient’s respiratory capacity [106]. Neutrophil count and extracellular DNA can be used as a severity marker for the assessment of inflammation and lung disease severity in CF [107]. High levels of NET components such as MPO and NE enzymes available in CF sputum and BALF are responsible for the damage of airway epithelium and connective tissues that correlate with lung disease severity [96,108,109]. DNase treatments in CF patients diminished this damage and improved pulmonary function [110].

Asthma is a chronic heterogeneous airway inflammatory disorder with symptomatic features of periodic wheezing, coughing, and shortness of breath, leading to the deterioration of lung function [111]. Asthmatic inflammation was thought to be attributed mainly to eosinophil, but recent research suggested a greater proportion of neutrophilic involvement having worsened disease severity with poor treatment outcomes with traditional glucocorticoids [112]. Extracellular traps have been marked in the airway of atopic allergic asthmatic patients [113]. In asthmatic patients, BALF neutrophil count and IL-8 are reported as the most powerful biomarker to differentiate severe and moderate conditions [114]. IL-8 is a neutrophil chemoattractant and established agonist of NETosis. Again in asthmatic patients, plasma-activated platelet also rises [115], which is also responsible for the induction of NET formation [24].

### 2.3. NETs in Cardiovascular Diseases

Acute myocardial infarction (AMI) is a life-threatening condition that occurs due to the blockade of blood flow to the heart muscle because of thrombus formation occluding one or more coronary arteries, resulting tissue damage. Neutrophil recruitment has been mentioned to be involved in the development of atherothrombosis [116], coronary thrombi [117] and also used as the prediction of acute coronary events [118]. High levels of NETs are associated with severe coronary atherosclerosis patients [119]. NETs are detected at the advanced atherosclerotic lesion both in human and mice models [120,121], and the application of DNase in Ischemia-reperfusion injury mice models reduces reperfusion injury [122]. Histone H4 structural components of NET scaffold results in atheromatous plaque instability, and anti-histone H4 antibody implication results in plaque stabilization [123]. A report described that NET could be a potential diagnostic marker in atherosclerosis [124]. Tissue factor is a transmembrane protein that stimulates the coagulation process. NET-associated tissue factor induces thrombogenic potential through platelet activation and increased thrombin generation, resulting in myocardial infarction [125].

### 2.4. NETs in Liver Diseases

The liver is the main organ for the clearance of circulating DNA and histones from the body. In Ischemia reperfusion, neutrophils are identified to the site of liver injury, and expulsion of damage-associated molecular patterns (DAMPs) and extracellular histones from damaged hepatocytes were observed to worsen the hepatic injury through TLR-4 and TLR-9 [126]. These DAMPs activate neutrophils to release NETs, which intensifies sterile inflammatory liver injury [127]. Again, extracellular histones activate the nucleotide-binding domain, leucine-rich repeat-containing protein 3 inflammasome, which further contributes to liver injury [128]. LPS activated platelets, and the subsequent NET formation showed disturbed microcirculation and liver damage [24]. In a mouse model of ischemia-reperfusion injury, NET-mediated amplified inflammation and liver damage were found to be restored after DNase or PAD4 inhibitor treatment [126].

### 2.5. NETs in Kidney Diseases

The pathophysiology of acute kidney injury involves the renal tubular cell death and auto amplification loop of cell necrosis called necroinflammation [129,130]. Infections, trauma, toxins, and ischemia induce a huge neutrophil recruitment in the renal tubule, resulting in necrosis and apoptosis and release of DAMPs and alarmins. Extracellular-free or NET-bound or immune-complex-associated histones are potent mediators of renal epithelial cell necrosis [131,132] and induce further histone release that acts as DAMPs [131,133]. These DAMPs and other inflammatory mediators further activate neutrophils to release NETs, which accelerate more surrounding tissue injury. Histones and NETs enhance tubular necrosis and capillary injury [134], and this glomerular injury can be rescued by using PAD4 inhibitor and anti-histone antibodies [132,135].

### 2.6. NETs in COVID-19

COVID-19 is a highly contagious respiratory disease caused by SARS-CoV-2 virus and was declared a global pandemic by the World Health Organization [136]. A higher presence of NETs in COVID-19 patients has been marked [137,138]. Neutrophilia, immune dysfunction and hyper-inflammation are the clinical features of severe COVID-19 patients [139,140]. This hyper-inflammation in COVID-19 is attributed to NETs [141]. Excessive production of cytokines termed cytokine storm is the hallmark in the pathogenesis of COVID-19 with augmented plasma levels of CCL 2, IFNγ, IFNγ-inducible protein 10, G-CSF, CCL3, IL-1β, IL-2, IL-6, IL-7, IL-8, IL-10, IL-17, and TNF-α, leading to subsequent severe consequences like ALI, ARDS, pulmonary thrombosis, and MODS [142,143,144,145,146]. IFN-γ, TNF-α, IL-1β, and IL-8 are strong agonists of NET induction [21,22,138]. NETs have been reported to contribute to the damage of the alveoli and pulmonary endothelium and immunothrombosis in patients with severe progression of COVID-19 [146,147,148,149,150,151]. Inflammatory microvascular thrombi having NET components have been identified in the lungs, kidneys, and hearts of COVID-19 patients [152]. NET components such as genomic DNA and citrullinated histones have been marked to initiate coagulation in COVID-19 patients through thrombin generation, resulting in poor fibrinolysis and dropped anticoagulant factor by binding to factor XII. NET inhibition in COVID-19 patients is found to compensate for the NET-induced inflammation and thrombotic tissue damage linked to COVID-19 ARDS and death [146]. Ultimately, there is a negative correlation between increased levels of NETs and decreased survivability in COVID-19 patients [153,154]. 

### 2.7. NETs in Coagulopathy and Thrombotic Microangiopathy

Thrombosis is a principle cause of morbidity and mortality. Vascular occlusion represents deleterious effects of NETs. DIC is a heterogeneous group of disorder featured by widespread activation of intravascular coagulation [70,155]. The interaction between coagulation and innate immunity results in immunothrombosis, and NET-induced immunothrombosis plays an essential physical role in innate immunity by immobilizing and preventing the dissemination of invading pathogens. Nevertheless, exaggerated NETs can initiate abnormal thrombogenicity, leading to DIC. NETs initiate a coagulation cascade and high-flow circulating NETs aggregate and stick platelets to its own scaffold and form RBC-rich thrombus that serves as a template for thrombus formation and adhere to the vascular endothelium in microvasculature [156]. Electrostatic interaction of the negatively charged cells with positively charged histones in NETs probably contributes to constructing the scaffold. This scaffold is responsible for deep-vein thrombosis (DVT). In fact, this NET–platelet–thrombin axis promotes intravascular coagulation and microvascular dysfunction [157,158]. Moreover, circulating free DNA suppresses fibrinolysis either by accelerating the inactivation of tissue plasminogen activator by PAI-I [159] or by thickening the fibers of fibrin clots in combination with histones [160]. Again, there is a recommendation for NETs to be an important factor in excessive and unbalanced thrombin generation because of their early presence at the onset of DIC [80]. Histones also play a key role in the NET-mediated coagulopathy. H3 and H4 of NETs stimulate platelets and induce thrombotic reaction in mice models [161]. Again in an animal model, NET and its integral part citrullinated histones H3 was found in thrombi and its intravenous administration induced clot formation [162]. 

Thrombotic microangiopathy (TMA) is a life-threatening condition and causes massive microvascular thrombosis with thrombocytopenia, microangiopathic hemolytic anemia, and MODS [163]. A high level of serum DNA–histone complexes and MPO in TMA patients implies the involvement of NETs in disease severity [164]. In mice models, treatment with DNase and PAD4 inhibitor blocks DVT, which reflects the involvement of NETs as potential agonists of thrombosis [156,165]. NE is a NET component, and in an experimental model of NE-deficient mice, thrombosis was found to be ameliorated [166]. Furthermore, low levels of plasma of DNase I activity were reported in thrombotic patients, which also passively proves the involvement of NET in thrombosis [167].

### 2.8. NETs in Diabetes

Diabetes mellitus (DM) is an array of metabolic diseases with clinical features of hyperglycemic conditions that arise due to impairment in insulin action, secretion, or the two. Hyperglycemia is connected to priming neutrophils for oxidative burst and ROS generation. Neutrophils and NETs are involved in the pathogenesis of both insulin deficiency type 1 diabetes (T1DM) and insulin resistance type 2 (T2DM) and subsequent complications [168]. In T1DM patients with disease severity of less than 1 year, circulating protein levels and enzymatic activities of proteinase 3 (PR3), NE, and MPO-DNA are found significantly higher compared to healthy controls, which indicates the amplified NET formation [169]. In an experimental model of T1DM, inhibition of neutrophil function and NETosis ablate the progression of diabetes [170]. In T2DM patients, elevated levels of NETs, neutrophil elastase, mono- and oligonucleosomes and cf-DNA are observed compared to healthy donors [171,172]. NETs are also responsible in the pathogenesis of diabetes-induced complications. PAD4, an enzyme taking part in chromatin decondensation during NETosis cascade and citrullinated histones, were markedly elevated in both T1DM and T2DM, which implicated poor wound healing in both mice and human, and interestingly, PAD4 inhibition with Cl amidine and application of DNase restore the wound healing [173,174]. A case–control study reported that the elevated levels of circulating NET components, DNA-histones, and NE are associated with the development of diabetes-induced retinopathy [175]. NETs are also reported in the sera of diabetic neuropathy [176] and nephropathy [177] patients.

### 2.9. NETs in Autoimmune Diseases

NETs and histones are strongly associated with autoimmune diseases [178]. NET exposure in the presence of B cells like immune cells develops antibodies directed against self-nucleic acids and cytoplasmic proteins such as MPO and PR 3 [179]. Prolonged presence of NETs triggers the production of anti-neutrophil cytoplasmic antibodies and anti-nuclear antibodies and vice-versa, which promotes vasculitis and enhances the autoimmune response [180]. NETs have been implicated in numerous autoimmune diseases like systemic lupus erythematosus (SLE) and rheumatoid arthritis (RA) [181,182]. RA is a chronic, deleterious systemic autoinflammatory disease of the synovial joints, and SLE is a complex and heterogeneous disease with characteristic features of the systemic multi-organ inflammation. Elevated propensity of spontaneous NET release along with high ROS, MPO, and NE had been identified from isolated neutrophils in patients with RA and SLE compared to healthy controls along with higher nuclear translocation of PAD4 [182,183,184]. Moreover, increased levels of NET formation and NET remnants MPO-DNA are also reported in the serum of RA patients [185]. In a rat model of RA, PAD4 inhibition by chloramidine compensated the NET-induced inflammation and erosive changes [186]. Another RA model study reports the reduction of joint inflammation and erosion when NETs inhibition was implicated by monoclonal antibodies directed against citrullinated histones [187]. In SLE, slow degradation of NETs and complement activation contributes to the disease progression [188]. In a murine model of SLE, NET inhibition by DNase I constrains the development of anti-ssDNA and anti-histone antibodies [189], and PAD4 inhibition by chloramidine reduces SLE-induced subsequent vascular injury and organ damage [190].

### 2.10. NETs in Cancer

Inflammation is a trademark of cancer and recent emerging evidence identified the presence of neutrophils as infiltrating inflammatory immune cells with tumor [191,192]. Neutrophil plays a controversial dual pro- and anti-tumorigenic role in tumor biology [193,194]. Consequently, the effects of NETs with tumors are also found in two reverse regulatory ways: pro-tumor effects that enhance cancer cell proliferation, invasion and metastasis and anti-tumor effects that inhibit proliferation and invasion [195]. Liu et al. hypothesized that NET-mediated anti-tumor effect could be through tumor cell destruction or immune system triggers. A recent study suggests cytotoxic and antineoplastic aspects of NETs against tumor cells [196]. The study revealed that the interaction of NET with melanoma cells can block tumor cell migration and reduce cell viability. Further intense exploration is required to investigate the anti-tumor effects of NETs. In this section, we will highlight the pro-tumorigenic role of NETs. Highly elevated levels of NETs deposition are marked in some malignant cancers with Ewing sarcoma [197], Lewis lung carcinoma (LLC) [198], breast cancers [199,200], and lymphoma [201].

Although the detailed underlying mechanism is yet to be investigated, several published literature studies disclosed that NETs promote tumor development by the inhibition of apoptosis and proliferative effects and thus uphold tumor progression and metastasis [202,203]. Circulating tumor cells (CTC) are significantly involved in tumor metastasis. Interestingly, due to its sticky mesh-like architecture, NETs can entrap and adhere to CTC and take to adjacent areas, thus expanding tumor cell metastasis. In an animal model of sepsis, significant levels of NET-associated tumor cells were clearly pictured upon injection of LLC cells compared to the healthy group, and cancellation of neutrophils voids this effect by lowering CTC adhesion within the liver [204]. A group of researchers further reconfirmed that only NETs but not neutrophils are responsible for augmenting tumor metastasis [201,205]. Moreover, the interaction of NETs with tumor cells induces a severe cancerous phenotype in cancer cells [195].

NET components also play a significant role in tumor progression [206]. NE has been described to have pro-tumorigenic activity and is found to induce proliferation and migration of tumor cells in vitro [195]. Lung adenocarcinoma model showed increased proliferation of a tumor cell line when neutrophils are co-cultured with A549 cells, and this proliferation is nullified when co-cultured with NE−/− neutrophils [207]. This pro-tumorigenic role of NE is implicated by the degradation of insulin receptor substrate-1 and subsequent activation of phosphatidaylinositol-4,5-bisphosphate 3-kinase [207,208]. In addition, NE also induces the release of pro-tumor factors like transforming growth factor ɑ, vascular endothelial growth factor, and platelet-derived growth factor, facilitating interaction with their respective receptors and favoring tumor progression [209].

Matrix metalloproteinase 9 (MMP-9), an integral part of NETs, degrades extracellular matrix and supports tumor metastasis [5]. MMPs favor a pro-tumorigenic course through tumor cell proliferation, impairing apoptosis, increased angiogenesis, invasion, and metastasis [210,211,212]. In mice models, this invasiveness of tumor is reversed in MMP-9−/− mice compared to wild-type mice [211].

Cathepsin G aided metastasis by increasing angiogenesis and tumor dispersion [213]. In animal models of breast cancer, cathepsin G potentiates tumor aggregation in vasculature and form distal tumor emboli [214]. In patients with hepatocellular carcinoma, NETs-associated cathepsin G induces tumor cell invasion [215].

### 2.11. NETs in Preeclampsia

Preeclampsia is a pregnancy disorder with characteristics of inflammation, hypertension, kidney failure, and seizures and is a significant cause of maternal and neonatal mortality worldwide [216]. Placental micro-particle-induced neutrophil extracellular DNA lattices are found in preeclampsia [23]. A huge elevation of circulating cf-DNA is measured from the maternal plasma [217]. A published literature indicates inappropriate NET formation as the source of this cf-DNA [218]. Elevated levels of microparticles from preeclampsia patients could trigger NETs in isolated neutrophils through an IL-8-dependent manner [219]. Moreover, NET components have been detected by immunofluorescence staining in the intervillous space of preeclamptic placentae [23,220]. Thus exacerbated NETosis leads to placental tissue damage in preeclamptic patients.

### 2.12. NETs in Kawasaki Disease

Kawasaki disease (KD) is an acute multisystem vasculitis syndrome with a characteristic feature of febrile illness and mainly affects infants and children in developed countries [221,222]. EC activation and injury is observed in patients with KD [223]. Although the etiology and mechanism are not crystal clear, however, KD patients are very susceptible to acquiring coronary artery abnormalities and myocardial ischemia [224,225]. The endothelial glycocalyx is a carbohydrate-rich gel-like layer lining buildup of syndecan, hyaluronic acid, chondroitin sulfate, and heparan sulfate [225]. A recent report demonstrates circulating endothelial glycocalyx proteins syndecan-1 and hyaluronan as predictive biomarkers of coronary artery lesions in KD [226]. Interestingly, NETs also promote cardiovascular EC damages, suggesting that enhanced NET generation may participate in the pathogenesis of KD vasculitis [227]. Again, high levels of proinflammatory mediators including TNF-α, IL-1β, and IL-8 have been reported in KD patients which are a strong inducer of neutrophils to release NETs [22,222]. The activation of neutrophils respiratory burst assessed by flow cytometry assay using dihydrorhodamine is described in the acute phase of KD [228]. Isolated neutrophils from acute-phase KD patients show enhanced NET formation compared to healthy controls [229,230,231]. Moreover, the NE plasma level is also higher in acute-phase KD [232]. Thus, NET and NET components contribute to KD and subsequent squeals.

## 3. Evaluation of NETs Inhibition as Therapeutic Targets

NETs are generated from the matured activated neutrophils in response to a wide array of infectious microbial pathogens and sterile stimuli and act as an indispensable protective barrier of innate immunity. Expelled NETs entrap and protect the dissemination of invading microorganisms throughout the body either by killing or confining them in its web-like structure. In contrast, aberrant activation of neutrophils and excessive production of NETs exaggerate inflammatory response that is likely to contribute to different diseases including infectious and non-infectious. Thus, NETs act as a double-edged sword due to their dual controversial role. Albeit the microbial infection would be eradicated by using broad-spectrum antibiotics in ICU, the issue of stopping the NET-induced systemic effects has come forward. We suggest that NET should be inhibited because it is cytotoxic. Recent evidence suggests that inhibition of NETosis does not hamper the killing capacity of neutrophils other than NETs [233]. The development of blocking NETs is in progress (Table 1) [234]. Inhibition of NETs by DNase and other inhibitors postulates benefits in the context of thrombosis [235], ischemia reperfusion injury [236], SLE [237], CF [108], AMI, stroke, diabetes [238], and cancer [239]. In pathological conditions, redundant NETs are not only present at sites of infection or inflammation but also available in the bloodstream, termed circulating cell-free NETs. Removal of circulating NET and NET components could be another potential therapy to prevent inappropriate inflammation and improve remote organ damage in critically ill patients. Recently, our group reported the removal of circulating NET components, cf-DNA, MPO-DNA, and NE-DNA by direct hemoperfusion with a cartridge containing polymyxin B (PMX)-immobilized membrane (Toray, Japan) in sepsis [240].

The therapeutic target and the time of initiation of the therapies are important concerns in the NET situation. This is because NETs are beneficial for the host in the early phase of the infections and become detrimental and backfire in later stages. So, the fine-tuning of NET formation throughout the disease course would be the goal for the development of new NET-targeted therapies. Early detection of NETs is essential to start the therapy and perhaps a combined therapeutic approach; inhibition of stimulation and damaged tissue repair would be the best option for treatment.

## 4. Conclusions

From their discovery in 2004, NETs have long been recognized as a novel central mechanism of innate immunity for their beneficial physiological role in host defense. But their pathogenic role has attracted recent attention. Several factors including aberrant activation, dysregulation, and excessive generation determine their controversial role. The list of NET-implicated diseases is gradually expanding, ranging from autoimmune disorders to diabetes to cancer [173,237,242]. This study will provide comprehensive knowledge to researchers, scientists, and clinicians to better understand the role and impact of NETs on health and address therapeutic targets to treat NET-mediated diseases. Although new novel therapeutic strategies are evolving targeting NETs, still, further and more extensive research is needed to explore their implication in different diseases

## Figures and Tables

**Figure 1 ijms-24-15805-f001:**
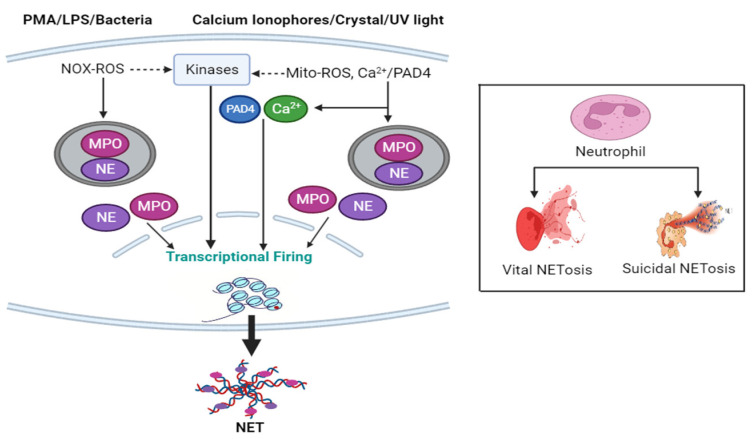
NET formation overview. NETosis occurs via two pathways. Cell-death-independent vital/non-lytic (faster) NETosis and cell-death-dependent suicidal/lytic (slower) NETosis. The NET formation mechanism is either Nox-dependent or Nox-independent depending on inducers. The cascade of events includes sets of kinases activation by Nox-ROS/Mito-ROS leading to transcriptional firing, MPO, NE, and PAD4 translocation from cytosol to the nucleus, facilitating chromatin decondensation, nuclear membrane disintegration, and finally NETs are expelled.

**Figure 2 ijms-24-15805-f002:**
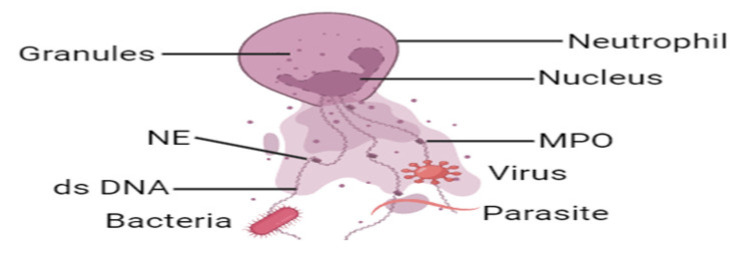
Physiological role of NETs. The web-like structure of NETs entraps intruding pathogens (bacteria, virus, fungi, and parasites) and prevents growth, proliferation, and spreading, to kill them and protect the host.

**Figure 3 ijms-24-15805-f003:**
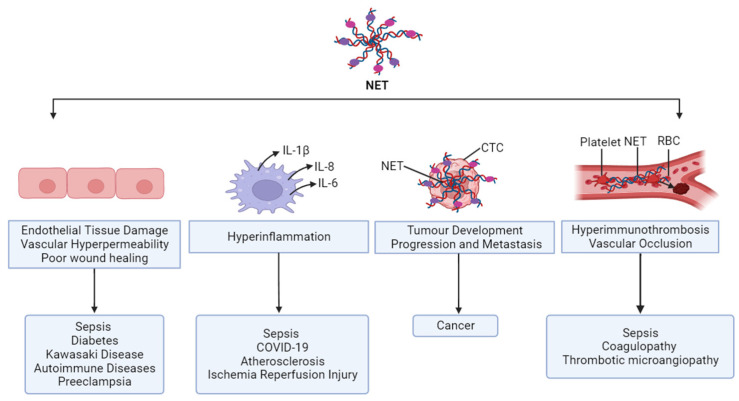
Controversial role of NETs. NET is a powerful weapon of innate immunity and provides protection to the host through clearing the invading pathogens. Aberrant NET formation causes pathology in numerous ways and is attributed to autoimmune, infectious, and non-infectious diseases.

**Table 1 ijms-24-15805-t001:** Inhibition of NETs as therapeutic approach.

References	Year	Results
Martinod et al. [235]	2014	NET inhibition by DNase I administration showed a protective role against thrombosis in an in vivo murine model of ischemic stroke, myocardial infarction, and DVT.
Brill et al. [166]	2012	Infusion of DNase I protected mice from DVT through inhibition of NETs.
De Meyer et al. [241]	2012	NET inhibition by DNase I improves ischemic stroke outcome in mice model.
Savchenko et al. [236]	2014	DNase I therapy against NETs in mice model exerts cardioprotective effects and improves cardiac contractile function.
Papayannopoulos et al. [108]	2011	DNase I therapy enhances sputum solubilization in CF patients
Wong et al. [238]	2018	Elevated NETs contribute to diabetes complications, and NET disruption by using DNase I therapy improves wound healing.
Patutina et al. [239]	2011	DNase I treatment inhibits tumor progression and dissemination.

## Data Availability

Not applicable.

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
