# Peer review of "Role of Neutrophil Extracellular Traps in Health and Disease Pathophysiology: Recent Insights and Advances"

_ijms, 2023, doi:10.3390/ijms242115805_

Round 1

Reviewer 1 Report

Comments and Suggestions for Authors

The review is a brief summary of literature data on the role of neutrophil granulocytes in defence against pathogens and in the development of a number of diseases. The primary attention is paid to NETs formation as the main way of realising the effector function of neutrophils.

In general, hundreds of articles have been written on this topic. For each subparagraph of this review, there are 2-3 "real" full-fledged reviews published in highly rated journals. The authors should explain their purpose in writing this review, its novelty, and for whom it is intended.

The review is titled " Neutrophil Extracellular Traps in Health and Disease  Pathophysiology: Recent Advances and Future Insights". The "Future Insights" are not disclosed.

Reviewer 2 Report

Comments and Suggestions for Authors

The review by Islam and Takeyama described the role of NET in different pathologies and its implication as putative therapeutic target. The review is interesting and quit well organized, even though I would address some suggestions. There are areas where the paper could be improved by clarity and dept.

The English used is generally good, but I recommend a proofreading by a native English speaker to improve the clarity and readability of the paper.

Line 79: typo -> “light” is missing an “l”

When you mention the name of bacteria, please, use italics.

I would suggest removing “etc” along the text. You are using it a lot, and from my point of view, it is not scientific at all. Replace or remove.

Figure 1: I suggest enlarging characters and as well as add something to the figure. In the present form it is not very explicative and does not add much to the review. I would also suggest adding, if possible, figures/tables in order to make the review easier for the readers and more attractive.

Lines 175-177: very long and confusing sentence. Crop and rephrase. You are repeating the same words too many times.

Lines 223-224: The introduction of cystic fibrosis is not precise and accurate. Please rephrase and cite some paper.

Line 225: CF trans-membrane conductance regulator add the acronym in brackets -> (CFTR).

Line 226: It is not the gene which controls the balance between the ions -> the osmotic balance is controlled by the protein CFTR encoded by the CFTR gene. Correct and add references to support the statement.

Lines 229-231: “and chronic” what is it referred to?

Lines 289-290: Please cite the OMS declaration of March 2020.

Lines 354 + 357: Please be consistent -> use on between PAD 4 or PAD4 along the manuscript.

I would also suggest adding the author point of view regarding the NET role and its possible implications as therapeutic target. I think it is an important point which would add value to the current work.

Comments on the Quality of English Language

The English used is generally good, but I recommend a proofreading by a native English speaker to improve the clarity and readability of the paper.

Round 2

Reviewer 2 Report

Comments and Suggestions for Authors

The  quality of the manuscript have been very much improved. I think the manuscript  should be accepted in the present form.
